



# biospheremetrics v1.0.1: An R package to calculate two complementary terrestrial biosphere integrity indicators: human colonization of the biosphere (BioCol) and risk of ecosystem destabilization (EcoRisk)

Fabian Stenzel[1], Johanna Braun[1], Jannes Breier[1], Karlheinz Erb[2], Dieter Gerten[1,3,4], Jens Heinke[1], Sarah Matej[2], Sebastian Ostberg[1], Sibyll Schaphoff[1], and Wolfgang Lucht[1,3,4]

[1]Potsdam Institute for Climate Impact Research (PIK), Member of the Leibniz Association, P.O. Box 60 12 03, D-14412 Potsdam, Germany
[2]Institute of Social Ecology, University of Natural Resources and Life Sciences, Vienna (BOKU), Schottenfeldgasse 29, 1070 Vienna, Austria
[3]Humboldt-Universität zu Berlin, Department of Geography, Unter den Linden 6, D-10099 Berlin, Germany
[4]Integrative Research Institute on Transformations of Human-Environment Systems, Unter den Linden 6, D-10099 Berlin, Germany

**Correspondence:** Fabian Stenzel (stenzel@pik-potsdam.de)

**Abstract.** Ecosystems are under multiple stressors and impacts can be measured with multiple variables. Humans have altered mass and energy flows of basically all ecosystems on Earth towards dangerous levels. However, integrating the data and synthesizing conclusions is becoming more and more complicated. Here we present an automated and easy to apply R package to assess terrestrial biosphere integrity which combines 2 complementary metrics:

The BioCol metric quantifies the human colonization pressure exerted on the biosphere through alteration and extraction (appropriation) of net primary productivity, whereas the EcoRisk metric quantifies biogeochemical and vegetation structural changes as a proxy for the risk of ecosystem destabilization.

Applied to simulations with the dynamic global vegetation model LPJmL5 for 1500-2016, we find that presently (period 2007-2016), large regions show modification and extraction of >25% of the preindustrial potential net primary production,

leading to drastic alterations in key ecosystem properties and suggesting a high risk for ecosystem destabilization. In consequence of these dynamics, EcoRisk shows particularly high values in regions with intense land use and deforestation, but also in regions prone to impacts of climate change such as the arctic and boreal zone.

The metrics presented here enable global-scale, spatially explicit evaluation of historical and future states of the biosphere and are designed for use by the wider scientific community, not only limited to assessing biosphere integrity, but also to

benchmark model performance.

The package will be maintained on GitHub and through that we encourage application also to other models and data sets.



**Table 1.** Abbreviations

| | |
|---|---|
| BFT | bioenergy functional type |
| CFT | crop functional type |
| GCT | ground cover type |
| HANPP | human appropriation of NPP |
| LPJmL | Lund-Potsdam-Jena managed Land (a dynamic global vegetation model) |
| BioCol | metric for human biomass colonization pressure |
| EcoRisk | metric for risk of ecosystem destabilization |
| NPP | net primary production |
| PFT | plant functional type |

# 1 Introduction

Earth system stability relies on functioning ecosystems (McKay et al., 2022), providing e.g. carbon sequestration, moisture recycling, and resilience to disturbances/disruptions (Friedlingstein et al., 2022; Aragão, 2012; Oliver et al., 2015). The global
status and functioning of ecosystems is being evaluated using a range of approaches, including assessments of human footprint (e.g. Venter et al. 2016), indicators based on empirically collected biodiversity data (e.g. Hudson et al. 2017; Newbold et al. 2016), or ecological computer models (e.g. Sakschewski et al. 2015).

Ecosystems depend on photosynthesis as major energy source, producing the primary biomass that is the foundation of almost all food-webs. During the neolithic revolution, humanity decoupled its biomass demand from the natural cycle, which
led to large-scale modification of Earth's surface (Weisdorf, 2005). Today more than 75% of the ice-free land area of the Earth is affected by human use (Watson et al., 2018; Arneth et al., 2019). However, the level of management and appropriation differs widely from extensive (e.g. occasionally livestock-grazed steppes or extensively used forests) to intensive (e.g. machine-aided agriculture with high mineral fertilizer use and irrigation). The aggregate effect of land use on NPP, i.e. altered productivity and removed biomass due to agricultural and forestry harvest, are often referred to as the human appropriation of NPP (HANPP)
(Haberl et al., 2004; Vitousek et al., 1986; Rojstaczer et al., 2001; Imhoff et al., 2004). Utilizing this concept, it was shown that humanity has doubled its impact during the 20th century and thereby substantially decreased the net primary production (NPP) remaining in ecosystems (Krausmann et al., 2013; Kastner et al., 2022).

As one consequence of land use, only 40% of remaining forests are still characterized by high ecosystem integrity (Grantham et al., 2020). Globally only 18.6% of highly intact habitats are currently protected (Mokany et al., 2020a), while modeling
pressure-impact relationships suggest an increasing mean species abundance loss until 2050, even under optimistic scenarios with decreasing land use (Schipper et al., 2020; Williams et al., 2021). Over the course of this century, the dilemma of negative effects from either climate change or climate change mitigation via large-scale biomass plantations highlights the need to integrate biodiversity also in planning for negative emission technologies (Hof et al., 2018).



To prevent further degradation of the biosphere and to reverse current loss of nature, land use scenarios should take into account the regional risk for ecosystem destabilization (Rockström et al., 2021; Obura et al., 2022). As ecosystem destabilization we understand a severe change in ecosystem functioning, resulting in e.g. a decline in carbon sequestration, species composition, or water provisioning. Shifts in biogeochemical conditions can act as a proxy for this risk, based on the argument that substantial changes in either basic biogeochemical properties or vegetation composition are likely to imply far-reaching, potentially self-amplifying transformations in the underlying system characteristics, food chains and species composition (Heyder et al., 2011). The $\Gamma$-metric (Gamma), proposed by Heyder et al. (2011) represents such a metric, which has been used to separate the historical drivers of ecosystem change (Ostberg et al., 2015), compare the effects of climate warming and land use under future climate scenarios (Ostberg et al., 2018), and find temperature thresholds above which severe change is to be expected with high probability (Ostberg et al., 2013). Additionally, it has been applied component-wise to model outputs from the ISIMIP fast track ensemble, indicating an increasing area under risk of severe ecosystem change with rise of global mean temperature (Warszawski et al., 2013).

However HANPP cannot be modelled throughout, because it is based on census statistics and inventory data - modelling approaches have not been existing so far. On the other hand, the original definition of $\Gamma$ did not include nitrogen variables and the code to compute it was hardly accessible to the scientific public.

We therefore propose an easy to apply R package with two complementary biosphere metrics, BioCol and EcoRisk, building on the existing indicators HANPP and $\Gamma$ (Haberl et al., 2004; Heyder et al., 2011), which are here exemplarily calculated and evaluated based on simulations with the global vegetation model LPJmL5 (von Bloh et al., 2018).

BioCol quantifies the human colonization pressure on the biosphere through extraction of biomass and prevention of natural NPP by photosynthesis. The metric basically follows the HANPP approach by Haberl et al. (2007), who spatially explicitly sum extracted and inhibited biomass amounts based on biomass inventory data and compare them to potential NPP in a counterfactual world without human land use. We replace biomass inventory data with the corresponding LPJmL5 model outputs and thereby add the possibility to compute BioCol also from deep historical and future simulations.

EcoRisk illustrates state shifts of ecosystems as a result of land, water, and fertilizer use, as well as climate change based on the $\Gamma$-Metric (Heyder et al., 2011; Ostberg et al., 2015). It quantifies on the scale of 0 (no change) to 1 (very strong change) the dissimilarity of an ecosystem state from a reference condition and comprises 4 subcomponents (vegetation structure, local change, global importance, ecosystem balance) that are aggregated as a multidimensional proxy for the risk of biosphere destabilization. We follow the original publications, but (additionally to water and carbon) now include nitrogen fluxes and pools. The addition of nitrogen variables fills a major gap, because nitrogen limitation or surplus is a key determinant for plant growth and the overall ecosystem status.

## 2 Material and methods

In this section, we detail the calculation of both metrics BioCol and EcoRisk, as well as our biome classification for spatial aggregation of results, and the relevant specifics of the vegetation model LPJmL5.





## 2.1 BioCol

We define BioCol (see Figure 1) as the flow of biomass (in terms of NPP) that is intentionally extracted in the form of crop, residue, and other biomass harvests ($NPP_{harv}$) plus the inhibited natural biomass production as a result of land use changes as well as management and differences in fires ($NPP_{luc}$). $NPP_{luc}$ is calculated as the difference between potential natural NPP (i.e. the NPP that would prevail without land use but with current climate (Haberl et al., 2014) – $NPP_{pot}$) and the actual biomass production ($NPP_{act}$), both calculated under the same climate:

$$NPP_{luc} = NPP_{pot} - NPP_{act} \qquad (1)$$

Using time series of $NPP_{pot}$ and $NPP_{act}$ allows to remove the climate change (e.g. $CO_2$ fertilization) effect in the component $NPP_{luc}$. $NPP_{luc}$ can become negative, if the actual NPP is higher than potential NPP (e.g. through management or land use legacy effects, especially on managed grasslands). Absolute HANPP and relative $BioCol$ are computed as

$$HANPP = NPP_{luc} + NPP_{harv} \qquad (2)$$

where $NPP_{harv}$ denotes the NPP withdrawn from ecosystems via harvest.

$$BioCol = HANPP/NPP_{ref} \qquad (3)$$

where $NPP_{ref}$ represents a reference NPP. In earlier applications, the respective $NPP_{pot}$ of each year has been used as reference (Kastner et al., 2022; Krausmann et al., 2013). In contrast, we here use $NPP_{ref}$, which refers to the $NPP_{pot}$ of the pre-industrial period (mean of 1550-1579), which is the same timeframe as for EcoRisk. $NPP_{harv}$ sums the corresponding LPJmL5 model outputs for harvested and extracted carbon from crop areas (including residues), second generation biomass plantations (not existing in historical period land use input) and grassland, as well as timber harvests from land use conversion, combined with external timber extraction from managed forests, and human induced fire carbon emissions. Residue harvest was assumed to contain 70% of the remaining above-ground biomass after harvest. Carbon extraction on grasslands includes carbon contained in dairy products and methane emissions plus respiration and is based on prescribed livestock densities (Heinke et al., 2023), which are calibrated to match grazing amounts for the year 2000 originally provided by Herrero et al. (2013) and modified by Heinke et al. (2020) (about 1.1 GtC/yr globally). Currently, LPJmL5 is not able to model managed forests, and also does not separate human induced and lightning induced fire carbon emissions. Therefore, timber harvest from managed forests is included as an external dataset from LUH2-v2h (Hurtt et al., 2020). Human induced fire carbon emissions can be included through external data on the human fire ignition fraction of total fire carbon emissions, based on fire models such as LPJmL-Spitfire (Thonicke et al., 2010). For this study, the fire carbon emissions are not included, since the updated LPJmL-Spitfire version is still being tested.

$$\text{human fire ignition fraction} = \frac{\text{human ignitions}}{\text{human ignitions} + \text{lightning ignitions}} \qquad (4)$$

$NPP_{pot}$, representative of the NPP under potential natural vegetation (under transient climate), is obtained from a model simulation without human land use, but otherwise identical settings to the run providing $NPP_{act}$. $BioCol$ or HANPP and



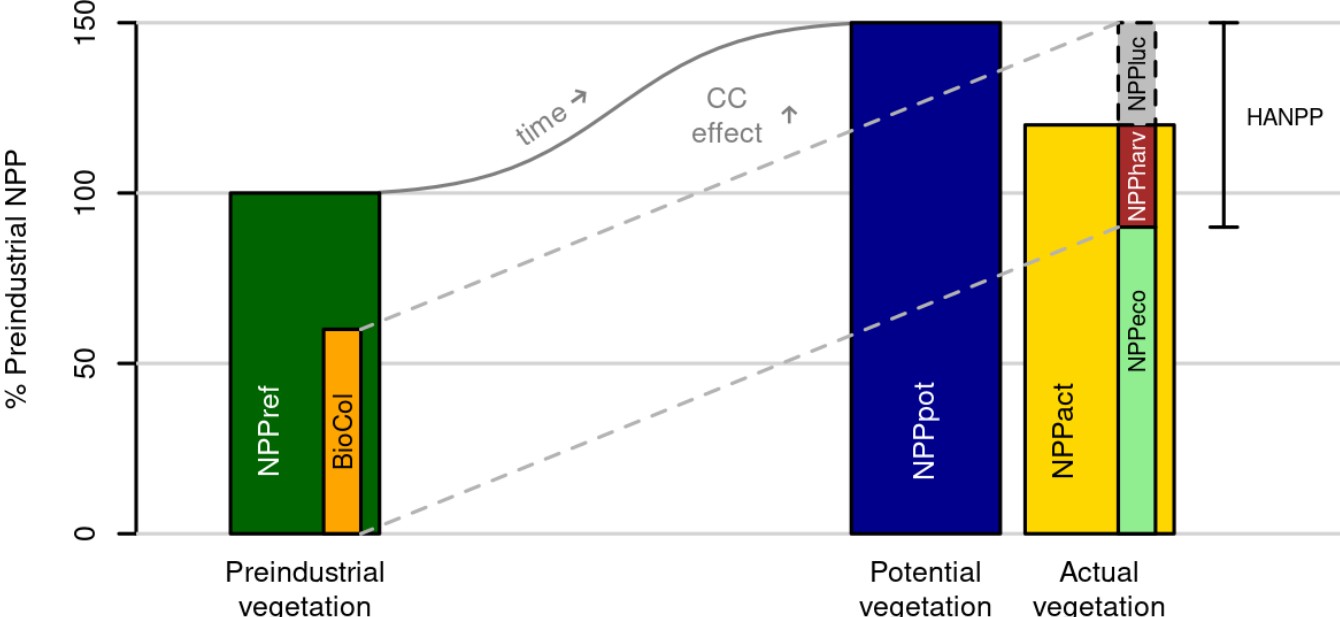

**Figure 1. Calculation scheme for BioCol.** The basis for our analysis is the preindustrial potential NPP (NPP$_{ref}$) assessed from 1550-1579. The effects of $CO_2$ fertilization of plants resulting from historical anthropogenic CO2 emissions, changes in atmospheric N deposition, and climate change lead to a net increase in NPP today (labeled as the "CC effect" in the biosphere), both for hypothetical "Potential vegetation" without human land use and the "Actual vegetation" including land use. HANPP is calculated as the sum of direct human biomass extraction (NPP$_{harv}$) and inhibited natural productivity through replacing natural vegetation with land use (NPP$_{luc}$ = NPP$_{pot}$ - NPP$_{act}$). BioCol is subsequently computed as the fraction of HANPP compared to NPP$_{ref}$.

sub-components are available as spatially explicit values per grid cell for every timestep, but also as global sums over time, or aggregated per biome or world region. When aggregating grid-cell values of BioCol, negative values can be treated as such, reducing the overall pressure, or the absolute values of all grid-cells can be summed up (which is mainly used in this analysis). For further details on the LPJmL5 simulation setup, see subsection 2.5.

## 2.2 EcoRisk

EcoRisk is computed as the average of four subcomponents: vegetation structure (vs), local change (lc), global importance (gi), and ecosystem balance (eb), each on the same scale of 0 to 1 and internally scaled with the respective change-to-variability ratio $S(x, \sigma_x)$:

$$\text{EcoRisk} = \frac{vs + lc + gi + eb}{4} = \frac{V \cdot S(V, \sigma_V) + l \cdot S(l, \sigma_l) + g \cdot S(g, \sigma_g) + e \cdot S(e, \sigma_e)}{4} \tag{5}$$

The unscaled **vegetation structure** component $V$ is evaluated based on the dissimilarity of the vegetation composition of the whole grid cell for both natural and managed areas based on Sykes et al. (1999); Heyder et al. (2011); Ostberg et al. (2018). For this, LPJmL5 provides outputs of the ground area covered by natural and cultivated plant types. Changes in vegetation structure



are computed between ecosystem state $i$ and $j$ with respect to the total area ($G$) of ground cover type (GCT) $k$ (tree, grass, or barren), further detailed by the differences between the plant functional type (PFT) ($p$) specific area ($A$) regarding attribute $l$ (evergreenness, needleleavedness, tropicalness, borealness, naturalness). For PFT specific attributes ($a$) see Table 2. Barren is defined without subcategories. The attribute specific weighting factor $\omega_{kl}$ is per default set to 0.2 (= "equal" weighting). Alternatively, attribute specific weights as in Ostberg et al. (2018) can be applied.

$$V(i,j) = 1 - \sum_k \left\{ \underbrace{\min(G_{ik}, G_{jk})}_{total\ area\ of\ GCT\ k} \cdot \left[ 1 - \underbrace{\sum_l \left( \omega_{kl} \left| \sum_p (A_{iklp} \cdot a_{klp}) - \sum_p (A_{jklp} \cdot a_{klp}) \right| \right)}_{specific\ area\ difference\ regarding\ attribute\ l} \right] \right\} \quad (6)$$

The remaining components $l$, $g$, and $e$ are computed from two ecosystem state vectors $\boldsymbol{s}_1$ (reference state) and $\boldsymbol{s}_2$ (changed state), composed of 30 year averages of biogeochemical variables $v_{1,i}$ and $v_{2,i}$, with $i = [1,...,n]$.

$$\boldsymbol{s}_1 = \begin{pmatrix} v_{1,1} \\ \vdots \\ v_{n,1} \end{pmatrix}, \boldsymbol{s}_2 = \begin{pmatrix} v_{1,2} \\ \vdots \\ v_{n,2} \end{pmatrix} \quad (7)$$

They represent the major process cycles on the grid cell level determining plant growth: carbon, nitrogen, and water. The
system described by the metric is the terrestrial land system, which has pools as well as in- and outfluxes across the system boundary. For carbon and nitrogen, stocks are aggregated into a vegetation and soil pool, for water there is only a soil pool. In total this gives 11 dimensions, plus one dimension reserved for other relevant processes (for LPJmL filled with the fire fraction) (Table 3). The user can define which model output variables constitute each process dimension. Hereby it is important, to know the model specific details to keep consistency. If possible, output variables should use per-area units because of the per area
weighting for the "global importance" component.

**Local change** describes changes compared to the local reference state (e.g. the preindustrial time) as the magnitude change of the difference vector of the biogeochemical properties (how strongly the local conditions have changed). For this, the state variables are normalized with the local values of the reference state:

$$\boldsymbol{s}_{l_1} = \begin{pmatrix} 1 \\ \vdots \\ 1 \end{pmatrix}, \ \boldsymbol{s}_{l_2} = \begin{pmatrix} v_{1,l} \\ \vdots \\ v_{n,l} \end{pmatrix} \quad (8)$$

with

$$v_{i,l} = \frac{v_{i,2}}{v_{i,1}}, \text{ for } v_{i,1} \neq 0. \quad (9)$$

For variables, which are 0 in both vectors, both are set to 1, resulting in no change. If only the reference case is 0, the unscaled values are used for both vectors.

**Global importance**, in contrast, puts these local changes in perspective to the global mean reference condition, taking
into account that even moderate changes on the local scale may feed back to larger scales if large enough in absolute terms.



**Table 2.** PFT specific attributes $a_{klp}$ for GCTs tree and grass.

| trees | | | | | |
|---|---|---|---|---|---|
| | evergreenness | needleleavedness | tropicalness | borealness | naturalness |
| Tropical broadleaved evergreen | 1 | 0 | 1 | 0 | 1 |
| Tropical broadleaved raingreen | 0 | 0 | 1 | 0 | 1 |
| Temperate needleleaved evergreen | 1 | 1 | 0 | 0 | 1 |
| Temperate broadleaved evergreen | 1 | 0 | 0 | 0 | 1 |
| Temperate broadleaved summergreen | 0 | 0 | 0 | 0 | 1 |
| Boreal needleleaved evergreen | 1 | 1 | 0 | 1 | 1 |
| Boreal broadleaved summergreen | 0 | 0 | 0 | 1 | 1 |
| Boreal needleleaved summergreen | 0 | 1 | 0 | 1 | 1 |
| Tropical bioenergy | 1 | 0 | 1 | 0 | 0 |
| Temperate bioenergy | 0 | 0 | 0 | 0 | 0 |

| grasses/crops | | | |
|---|---|---|---|
| | tropicalness | borealness | naturalness |
| C4 grass tropic | 1 | 0 | 1 |
| C3 grass temperate | 0 | 0 | 1 |
| C3 grass polar | 0 | 1 | 1 |
| Temperate cereals | 0 | 0 | 0 |
| Rice | 1 | 0 | 0 |
| Maize | 1 | 0 | 0 |
| Tropical cereals | 1 | 0 | 0 |
| Pulses | 0.5 | 0 | 0 |
| Temperate roots | 0 | 0 | 0 |
| Tropical roots | 1 | 0 | 0 |
| Sunflower | 0.5 | 0 | 0 |
| Soybean | 1 | 0 | 0 |
| Groundnut | 1 | 0 | 0 |
| Rapeseed | 0.5 | 0 | 0 |
| Sugarcane | 1 | 0 | 0 |
| Others | 0.5 | 0 | 0 |
| Managed grass | dyn* | dyn* | 0 |
| Bioenergy grass | 1 | 0 | 0 |
| Grass under bioenergy trees | dyn* | dyn* | 0 |

* dynamic share due to climate specific grass mix



**Table 3.** Processes and associated variables describing the land system and their aggregation from LPJmL5 outputs.

| Process description | LPJmL5 variable(s) | Relevance for ecosystem |
|---|---|---|
| **vegetation structure** | | |
| surface area covered | Natural vegetation and bioenergy: foliage projected cover | available plant groups |
| by respective plant groups | Crops: land use fraction | and their surface coverage |
| **carbon pools** | | |
| vegetation carbon pool | vegetation carbon | carbon stored in vegetation |
| soil carbon pool | soil carbon + litter carbon | carbon stored in soil |
| **carbon fluxes** | | |
| carbon influx | GPP | source of exergy for ecosystem |
| carbon outflux | autotrophic respiration + heterotrophic respiration | carbon losses |
| | + fire carbon emissions + harvested carbon (crops+residues) | |
| **nitrogen pools** | | |
| vegetation nitrogen pool | vegetation nitrogen | nitrogen stored in vegetation |
| soil mineral nitrogen pool | soil $NH_4^+$ + soil $NO_3^-$ | reactive nitrogen stored in soil |
| **nitrogen fluxes** | | |
| nitrogen influx | biological nitrogen fixation + fertilizer nitrogen input | nitrogen entering the system |
| | + manure nitrogen input + atmospheric nitrogen deposition | |
| nitrogen outflux | harvested nitrogen (crops+residues) + nitrogen leaching to surface water | nitrogen leaving the system |
| | + $N_2O$ emissions from denitrification and nitrification | |
| | + $N_2$ emissions + nitrogen volatilization + fire nitrogen emissions | |
| **water pool** | | |
| soil water pool | root zone soil moisture | available green water |
| **water fluxes** | | |
| water influx | precipitation + irrigation | sources of water |
| water outflux | plant transpiration + soil evaporation + interception + runoff | sinks of water |
| **other** | | |
| other processes | fire frequency | other ecosystem-relevant processes |



Therefore, the state vectors are normalized with the global, spatially averaged reference mean value $\overline{v_{i,refg}}$:

$$\boldsymbol{s}_{g_1} = \begin{pmatrix} v_{1,g,1} \\ \vdots \\ v_{n,g,1} \end{pmatrix}, \ \boldsymbol{s}_{g_2} = \begin{pmatrix} v_{1,g,2} \\ \vdots \\ v_{n,g,2} \end{pmatrix} \tag{10}$$

with

$$v_{i,g,t} = \frac{v_{i,t}}{\overline{v_{i,refg}}}, \text{ for } \overline{v_{i,refg}} = \frac{1}{\sum a_p} \sum_{p=1}^{z} a_p v_{i,p} \neq 0. \tag{11}$$

for cells $p = 1, ..., z$ with cell area $a_p$. If the global mean reference state is 0, instead the mean scenario state is used for scaling. If both are 0, both vectors are set to 1, as for local change. Afterwards both state vectors for global importance and local change are multiplied by $\frac{1}{\sqrt{n}}$ to scale it down according to the number of variables (EcoRisk can also be computed for simulations without nitrogen, and thus missing the corresponding variables). The difference between the two states is now characterized by the length of the difference vector between them, which for local change and global importance are defined as:

$\quad d_l = |\boldsymbol{s}_{l_2} - \boldsymbol{s}_{l_1}|, \ d_g = |\boldsymbol{s}_{g_2} - \boldsymbol{s}_{g_1}| \tag{12}$

**Ecosystem balance** quantifies shifts in the relative magnitude of biogeochemical properties with respect to each other as an indicator for qualitative changes in the balance of dynamic processes, which may signal a breakdown of ecological functioning (Ostberg et al., 2018). It is calculated from the angle between the two state vectors with local normalization (as for local change):

$\quad b' = 1 - cos(\alpha) = 1 - \frac{\boldsymbol{s}_{l_1} \cdot \boldsymbol{s}_{l_2}}{|\boldsymbol{s}_{l_1}||\boldsymbol{s}_{l_2}|} \tag{13}$

$b'$ is scaled to a range between 0 and 1 assigning a value of 1 if the angle between state vectors is larger than $60°$:

$$e = \begin{cases} b' \cdot 2 & \text{if } \alpha \leq 60°, \\ 1, & \text{otherwise} \end{cases} \tag{14}$$

Values for metric components l and g are derived by scaling $d_l$ and $d_g$ to a range between 0 and 1 using the following sigmoid transformation function T:

$\quad l = T(d_l), \ g = T(d_g) \tag{15}$

$$T(x) = A + \frac{1 - A}{1 + e^{-6(x-0.5)}} \tag{16}$$

with $A = -\frac{1}{e^3}$.

The year-to-year variability is accounted for by the **change-to-variability ratio**. This variability factor is based on the

standard deviation ($\sigma_x$) of the component in the 30 year reference period, based on the assumption that ecosystems are adapted



to the variability they are regularly exposed to but may be vulnerable if it is exceeded. The change-to-variability ratio $S(x, \sigma_x)$ for components $x \in (V, l, g, e)$ is calculated as

$$S(x, \sigma_x) = \frac{1}{1 + e^{-4(\frac{x}{\sigma_x} - 2)}} \tag{17}$$

with $\sigma_x$ the interannual standard deviation of x under reference conditions.

For the specific status of variables describing the same process (e.g. carbon pools, or water fluxes), the change metric is also evaluated only for those variables, per default as $(lc + gi + eb)/3$, but for compatibility with earlier applications, it is possible to only use $lc$.

### 2.3    Comparison to other indicators

To contextualize EcoRisk and BioCol with other biosphere integrity indicators, we transformed a set of 7 widely used biosphere
integrity indicators (Table 4) to the interval [0,1] (with 0 meaning high integrity/no pressure/low risk) and show them alongside EcoRisk and BioCol, together with scatterplots of EcoRisk and BioCol against the average value across all indicators. We additionally extracted thresholds for each indicator from the literature, that demarcate the transition between the low and high risk zone (sources listed in Table 4). With these, we calculated the number of indicators per grid cell, that show up as transgressed.

### 2.4    Biome classification

To aggregate BioCol and EcoRisk from grid cells to the biome level (to also enable analysis for these larger-scale ecological units), we updated the biome classification for LPJmL5, originally published by Ostberg et al. (2013). It is based on the vegetation structure and its fractional coverage per cell, tree specific leaf area index, temperature, and the carbon stored in vegetation. Cells are classified primarily based on the total tree cover. Thresholds of 60%, 30% and 10% are chosen to demarcate the
boundaries between forests, woody savanna/woodland, savanna, and grassland based on the IGBP land cover classification system (Belward, 1996). The dominant tree or grass species then determines the type of biome. Compared to Ostberg et al. (2013), the thresholds can be adjusted and an additional differentiation between Tropical Rainforest and warm woody Savanna/Woodland can be either based on the tree leaf area index, or the vegetation carbon (in this paper, we use a tree leaf area index of 6 as threshold). Additionally, Montane Grassland can be differentiated from Arctic Tundra by elevation or latitude
(here, we use an elevation threshold of 1000m). Today's biomes are classified as shown in.

### 2.5    LPJmL5

We employ the dynamic global vegetation model LPJmL 5.7 (Schaphoff et al., 2018; von Bloh et al., 2018), which can be run standalone (forced by climate inputs) or coupled into the Potsdam Earth Model POEM (Drüke et al., 2021).

LPJmL5 simulates key ecological and physiological processes such as photosynthesis, respiration, carbon allocation and
turnover for natural and managed vegetation based on historical data or future projections of land use, soil, nitrogen inputs, and climate conditions. The composition of natural vegetation, represented by 11 PFTs, dynamically develops based on climatic



**Table 4.** Sources for other biosphere integrity indicators and associated thresholds.

| Indicator | range | source | threshold | threshold source |
|---|---|---|---|---|
| GLASOD: degree of human induced soil degradation | 1-4 | Oldeman et al. (1990) | strong/extreme: >2 | Oldeman et al. (1990) |
| HF: human footprint | [0,50] | Venter et al. (2018) | high pressure: >6 | Venter et al. (2016) |
| BII: biodiversity intactness index | [0.5,1] | Newbold et al. (2016) | BII target: >0.8 | Soergel et al. (2021) |
| intactness: GLOBIOM 2015 MSA https://portal.geobon.org | [0,1] | Schipper et al. (2020) | degraded: <0.6 | visual inspection |
| FLII: Forest Landscape Intactness Index | [0,10] | Grantham et al. (2020) | low integrity: 0-6 | Grantham et al. (2020) |
| CI: contextual intactness | [0,1] | Mokany et al. (2020a) | high-value: >0.5 | Mokany et al. (2020b) |
| CoE: Convergence of Evidence from World Atlas of Desertification wad.jrc.ec.europa.eu/geoportal | [0,10] | Cherlet et al. (2018) | many: >7 | Cherlet et al. (2018) |

parameters, growth efficiency, competition and fire disturbance (Sitch et al., 2003), while agricultural crop composition is prescribed considering 12 crop functional types (CFTs) (Bondeau et al., 2007), grassland/pastures (Heinke et al., 2023), and three second-generation bioenergy crop functional types (BFTs) (Beringer et al., 2011). The remaining group of "other" crops

is simulated as temperate wheat or tropical maize, depending on the latitude. CFT specific sowing dates are dynamically calculated based on optimal season length and fixed after the year 2000 (Waha et al., 2012). Local runoff is routed through a global network of water bodies and river channels as accumulated discharge (Gerten et al., 2004). Irrigated crop area is prescribed, but irrigation requirements and water withdrawals are dynamically simulated based on the soil water deficit and available renewable water in lakes, rivers, reservoirs as well as neighboring cells with available water (Jägermeyr et al., 2015).

Different agricultural management practices and their impacts on soil processes and yields are simulated, including tillage, manure, and fertilizer application as well as optional growth of off season cover crops (Lutz et al., 2019; Porwollik et al., 2022). Fluxes and stocks of carbon, nitrogen and water are by default resolved on 0.5°x0.5° spatial and daily time scale. For this study all outputs are aggregated per year.

Output from LPJmL5 to compute BioCol and EcoRisk is read and processed using the R package lpjmlkit (Breier et al., 210 2023).





We ran the model with a fused climate input from the ISIMIP project (GSWP3-W5E5). It combines daily GSWP3 data from 1901 to 1978 with a bias-adjusted version of ERA5 (W5E5, adjusted to better match CRU and GPCC) from 1979 to 2016 (Kim, 2017; Lange, 2019). Manure, fertilizer, and crop specific cultivation areas from 1500 to 2018 are taken from a new hybrid dataset (Ostberg et al., 2023). Fallow land was included in the CFT class "others", while the distribution of irrigated area into

irrigation systems is based on Jägermeyr et al. (2015). Livestock densities on grasslands were determined based on estimates of grazing levels at both regional and production system levels reported by Herrero et al. (2013), using a set of simulation presented by Heinke et al. (2023). Tillage input is based on Lutz et al. (2019). For simulation years before 1901, the first 30 years of the climate input are randomly recycled.

## 3 Results

The results of BioCol are (except for forest harvest) strictly model-based and are in general well in line with the existing data-based assessments (Kastner et al., 2022; Krausmann et al., 2013). The global sum of HANPP, based on LPJmL5 simulated outputs, increases from below 2 [2.5] GtC/yr before 1700 to more than 13 [15] GtC/yr today, depending on whether the absolute values are summed up for cells with negative values or not (Figure 2a - left y-axis). Over the course of the 20th century, relative values of BioCol thus approximately doubled from about 0.1 to over 0.2 (10% to over 20%) compared to the

potential preindustrial NPP$_{ref}$ of the 16th century (Figure 2a - right y-axis). Since 2000 the values increased further. When taking the absolute for negative cells, BioCol is approaching 0.25 (25%) in 2016 – meaning that almost a quarter of the total terrestrial preindustrial biomass production on Earth is rerouted to human use or inhibited compared to a world without humans.

The global relative BioCol pattern for the year 2000 (1995-2005 average) shows high values for areas of intense agricultural use (Figure 2b). The abundance of cells with negative BioCol values (higher productivity in the simulation with land use than in

the similar run with only potential natural vegetation) might be explained by the explicit simulation of agricultural management especially in regions with low natural NPP (e.g. irrigation in the Middle East) and potential legacy effects from earlier land use on now natural areas included in our land use dataset (e.g. in the Amazon, NPP upon regrowth of natural vegetation on abandoned agricultural areas is simulated to be higher than potential NPP in the equilibrium state; see Figure A1).

Isolated cells with very high absolute BioCol values (e.g. in the Boreal zone) are locations of reservoirs established before

the year 2005, with associated decline in NPP and thus high values of $NPP_{luc}$.

EcoRisk calculated between 1550-1579 and 1985-2016 shows large areas with values above 0.5 (Figure 3), mostly in regions with high land use intensity today. Land use change is reflected in the vegetation structural change component (vs), which resembles quite closely the current land use extent (see inlet). However the components building on changes in biochemical variables (lc, gi, eb) indicate a much larger extent of region with values > 0.9. Figure 4b and Figure A2 show that in many

regions changes in nitrogen fluxes are responsible for these strong changes.

Climatic changes are also reflected in EcoRisk. They are most visible in the components ecosystem balance (eb) and vs as bands of higher values along the frontier of boreal vegetation onset to the North in Canada. In Eurasia (and other land use free regions of the Arctic) this vegetation shift appears more patchy. The local change component (lc) indicates high values among





**Figure 2. (a)** Global BioCol and components over time. **(b)** map of the relative values for the year 2000 (average 1995-2005). Relative values for BioCol are expressed in comparison to the average NPP from 1550-1579 from a run without human land use.

especially vulnerable ecosystems with low absolute values (arctic, deserts), which do not show up in the global importance
(gi). gi primarily shows high values in regions with strong absolute changes, e.g. with intense land use, or loss of tropical







**Figure 3. (a)** Change in biochemical compositions computed by EcoRisk between 1550-1579 and 1985-2016. **(b)** current land use extent for reference. **(c-f)** EcoRisk components: vegetation structure change, local change, global importance, ecosystem balance

rainforests. Ostberg et al. (2018) provides a more detailed discussion of separate and combined effects of land use and climate change (however for a somewhat differently defined metric and using an earlier model version and different input database).

For the original Γ-metric, a threshold of 0.3 had been established, which indicates the transition from moderate to high risk of ecosystem destabilization. To determine whether this threshold is still valid for EcoRisk (given the adaptations to the





computation, particularly the inclusion of nitrogen flows and pools which show more variability), we performed two sets of
synthetic simulations. We assume that a meaningful threshold should meet the following two criteria: (i) it should be higher
than internal variability within biomes, but (ii) lower than the variability between distinct biomes, such that a simulated EcoRisk
above this threshold is indicative of changes equivalent to a shift in biome. For (i) we checked the homogeneity within forest
biomes, by computing EcoRisk with values from 1550-1579 between each cell of a biome and the average cell of this biome.
In all forest biomes, internal biome variability for at least half the cells is higher than 0.3 compared to the average biome cell
(Figure A3). Thus, values of < 0.5 could better describe most of the internal variation.

For (ii), we compared the average biome cells against each other by computing EcoRisk between the current states of each of
them. Most combinations show EcoRisk values > 0.3, when compared against each other (Figure A4), with some exceptions:
Boreal Needleleaved Evergreen Forest is classified as relatively similar to Temperate Broadleaved Deciduous Forest (0.26), a
fact that we cannot explain - and also complicated by the reverse direction showing a high value of 0.62. Tropical Rainforest is
classified as similar to Tropical Deciduous Forest. This can be partially explained through their similar locations and overlap
in species. Arctic Tundra and Montane Grassland are classified as similar. They are in fact the same biome, but at different
elevation. Comparably, Temperate Savanna, Temperate Woodland and Warm Grassland are classified as similar biomes, only
differentiated by total tree cover fraction and C3/C4 grass shares. There are further biomes which show intermediate values of
0.3 < EcoRisk < 0.5 when compared against each other, all partially explainable through compositional similarity or similar
average conditions.

However, Figure A4 also highlights that EcoRisk is not symmetric, mainly because of the normalization to different reference
conditions for local change and ecosystem balance. Strong directional discrepancies can for example be observed for the
difference between the three biomes "Temperate Needleleaved Evergreen Forest", "Temperate Broadleaved Deciduous Forest",
and "Boreal Needleleaved Evergreen Forest".

Aggregating EcoRisk grid cell values to the biome level, according to the biome classification yields three classes (Figure 4):
1) Those with a median EcoRisk < 0.3: "Tropical Rainforest", "Tropical Deciduous Forest", "Boreal Broadleaved Deciduous
Forest", "Boreal Needleleaved Deciduous Forest" (only few cells), "Arctic Tundra" - 2) those with 0.3 < median EcoRisk < 0.65:
"Boreal Needleleaved Evergreen Forest", "Warm Woodland", "Warm Savanna", "Warm Grassland", "Temperate Savanna",
"Temperate Grassland", "Montane Grassland", "Desert" - and 3) those with a median EcoRisk > 0.65: "Temperate Broadleaved
Deciduous Forest", "Temperate Needleleaved Evergreen Forest", "Temperate Woodland". The subcomponent local change is
generally the one with highest values (except for TrRF, TrDF, BoND) and nitrogen fluxes in all cases show stronger changes
than those for carbon and water.

Comparison of EcoRisk and BioCol with other biosphere integrity indicators (Table 4), shows similar trends despite high
scattering (Figure 5k/m). The maps highlight that the pattern of BioCol is more similar to that of other transgressed indicators
(Figure 5a-j), however this might stem from most of them being directly affected by land use. The additional benefit of EcoRisk
is that it also captures effects due to climate and deposition changes.



## 4   Discussion

We present a model-based indicator set that allows to assess the state of the biosphere. We show that presently (period 2007-
2016), large regions show modification and extraction of >25% of the preindustrial potential net primary production according
to the indicator BioCol, along with climatic changes leading to drastic alterations in key ecosystem properties and suggesting
a high risk for ecosystem destabilization according to the indicator EcoRisk.

Generally the indicators presented in this package can serve both as an analytic tool to assess biosphere integrity from
model simulations, or for means of benchmarking model performance (e.g. after new development). Therefore depending on
the context, the performance of the vegetation model is important. This paper primarily describes the methodology behind the
biospheremetrics package, with the application to model results only being secondary. LPJmL5 is currently in a recalibration
and validation phase, with major changes to code and key processes, following the implementation of tillage and the nitrogen
cycle. Further model development may particularly focus on an improved distribution of PFTs and thus biomes (see Harper
et al. 2023 for comparison) and the explicit simulation of multi-cropping, which has been neglected here. A better representation
of human induced fire emissions and including process based forest management will also be important, since the effects on
NPP are currently not considered.

The next step for us is thus to extend the biospheremetrics package to be compatible with other vegetation models (e.g.
utilizing outputs from the ISIMIP3 ensemble) and thus also allow for intermodel comparison.

As presented in the previous section, the addition of nitrogen fluxes leads to a strong increase in values for EcoRisk, when
comparing to earlier results of the $\Gamma$-Metric (Figure 3, Figure A2 and Ostberg et al. 2015, 2018). Relative changes in nitrogen
fluxes (Figure A5) as a result of nitrogen fertilizer and manure application are much stronger than those for carbon and water
fluxes and pools (Figure A6, Figure 4). This is intuitively plausible, however the question remains, whether the associated high
values for EcoRisk mainly resulting from the relative changes in nitrogen fluxes really correspond to a strongly increased risk
for ecosphere destabilization. Generally, a theory of how changes in different components (vs, lc, gi, eb) or classes of state
variables (e.g. water pools or nitrogen fluxes) can be translated to risk of ecosphere destabilization is lacking (and could be
different among components/classes). Theoretically, a weighted downscaling of any component would be possible, however
the literature base for such changes is currently lacking. We thus refrained from any weighting until further research is done on
this end.

The results for EcoRisk show a strong increase in the overall values. A new threshold between moderate and high risk
(replacing 0.3 in Ostberg et al. 2018, 2015; Heyder et al. 2011) would be as high as 0.48, when picking the "optimal" EcoRisk
threshold, balancing between forest biome internal homogeneity on the one and inter-biome dissimilarity on the other hand
(Figure A7).

Our way to calculate BioCol differs substantially from previous approaches for HANPP. First, our calculation is exclusively
based on model output, with the aim to also simulate future scenarios and historical periods outside of available biomass
inventory data. But we also use a different baseline, when comparing to the static preindustrial NPP and taking the absolute
of cells with negative HANPP values. Both is done deliberately in order to 1) not let ourselves get distracted from the fact





that global NPP is mainly rising because the biosphere is in a resilience response phase to increasing atmospheric $CO_2$ levels, meaning that this additional NPP is not ours to use, and 2) acknowledge that management driven NPP increases beyond the potential natural values (negative $NPP_{luc}$) usually go together with modification of the water, carbon, or nitrogen cycles stressing local ecosystems.

The underlying principle of BioCol, NPP appropriation, is a function of human demand, i.e. land use change, and biomass harvesting for food, fiber, fodder, bioenergy and bioeconomy. It is therefore closely associated with issues not just of Earth system stability but also of justice, access, and sustainable management of resources (see Gupta et al. 2023). The challenge is to maintain the productivity of the biosphere, i.e. the sustainably available NPP, while ensuring the stability of the biosphere, and beyond that, including climate and the ecosphere, i.e. the Earth system as used by humans.

Complementary, EcoRisk indicates where and when critical transitions in ecosystems occur (as a result of NPP appropriation, but also from climate change, or other environmental pollution). It is thus a very useful indicator to assess the risk of ecosystem destabilization today and also forward, depending on which pathway humanity will take in the future. We deliberately call it a risk metric, because the mathematical property of measuring a non-directional change for us directly translates to a proxy for the risk of ecosystem destabilization. Of course, there could be regions, where ecosystems can benefit from changing biogeochemical properties, however we would argue that when comparing to a long-term stable, land use free preindustrial reference situation (like the Holocene), most ecosystems would be at the risk of being thrown out of this equilibrium when presented with changing conditions. This choice of reference might need to be changed for application of EcoRisk to future climate stabilization scenarios with "Earth-System Stewardship" (Rockström et al., 2021; Steffen et al., 2018).

## 5 Conclusions

Humanity faces the challenge to stabilize the Earth System in the Anthropocene. This requires stabilizing the climate as well as a maintaining a resilient biosphere, which represents, according to Steffen et al. (2015), a key pillar of the Earth system. We here present two computable biosphere integrity indicators which allow to assess historical and future risk of biosphere destabilization. These are designed to complement other biosphere integrity indicators.

Human appropriation of NPP is a direct result of land-use changes, modulated by climatic changes, irrigation, fertilization and other management, leading to biosphere integrity loss that can be measured by EcoRisk. On the other hand, also natural NPP is modified by changes in climate, water availability, biosphere integrity, and human land-use. Thus, BioCol and EcoRisk are metrics which integrate multiple terrestrial planetary boundaries (Rockström et al., 2009; Steffen et al., 2015) and human drivers into two numbers, comparable to what global mean temperature does for the underlying, more complex climate processes. Therefore, BioCol and EcoRisk have the potential to be included as indicators in an updated planetary boundaries framework. While well-established as a concept, there so far has been no simulation model for assessing and projecting HANPP on the global scale, to for example analyze future scenarios not accessible through inventory data.

Both BioCol and EcoRisk are spatially explicit, process-based, computable metrics that can be aggregated over space and time. They can therefore serve as integrative meta-level proxies for biosphere integrity and Earth system stability and their



dynamic changes. The code used here for their calculation is distributed as an open source R-package and we invite external
contributions. With the next update, we plan to equip it to deal with netcdf files, a common data exchange format for spatial
data, which allows for future application to different models. Thereby, we hope to encourage others to join our quest to better
understand the role of ecosystems for Earth system stability and find ways of how to preserve it.

*Code and data availability.*  The package is available via github https://github.com/stenzelf/biospheremetrics, permanently archived via zen-
odo https://doi.org/10.5281/zenodo.10047951. LPJmL model code, simulation data, and scripts to generate the Figures are available on
Zenodo https://doi.org/10.5281/zenodo.10008051.



**Figure 4. (a)** Today's (1987-2016) biomes classified from vegetation structure, plant specific leaf area index, temperature and elevation in LPJmL5. **(b)** Change in biochemical compositions computed by EcoRisk between 1550-1579 and 1985-2016 as the median (Q10 and Q90 for whiskers) across the 16 most relevant biomes ("Temperate Broadleaved Evergreen Forest" basically does not establish – only 2 cells are classified as such – "Rocks and Ice" as well as "Water" skipped for lack of vegetation). For table with biome names and abbreviations see Table A1.




**Figure 5.** Contextualization of several indicators of biosphere integrity, transformed to the interval [0,1], with 0 meaning high integrity/no pressure/low risk. **(a)** GLASOD: human induced soil degradation (Oldeman et al., 1990), **(b)** HF: human footprint (Venter et al., 2016), **(c)** BII: biodiversity intactness index (Newbold et al., 2016), **(d)** intactness: GLOBIOM 2015 MSA (Schipper et al., 2020), **(e)** FLII: Forest Landscape Intactness Index (Grantham et al., 2020), **(f)** CI: contextual intactness (Mokany et al., 2020b), **(g)** CoE: Convergence of Evidence from World Atlas of Desertification (Cherlet et al., 2018). **(h)** number of previous 7 indicators per grid cell that show up as transgressed (see Table 4 for thresholds indicating transition between low and high risk zones), **(i)** EcoRisk and **(j)** BioCol **(l)** average of metrics a-g **(k)** scatterplot EcoRisk vs average, **(m)** scatterplot BioCol vs average.



**Table A1.** Names and abbreviations for all biomes according to our biome classification.

| Biome | Abbreviation |
| --- | --- |
| Tropical Rainforest | TrRF |
| Tropical Seasonal & Deciduous Forest | TrDF |
| Temperate Broadleaved Evergreen Forest | TeBE |
| Temperate Broadleaved Deciduous Forest | TeBD |
| Temperate Needleleaved Evergreen Forest | TeNE |
| Boreal Needleleaved Evergreen Forest | BoNE |
| Boreal Broadleaved Deciduous Forest | BoBD |
| Boreal Needleleaved Deciduous Forest | BoND |
| Warm Woody Savanna, Woodland & Shrubland | WaWo |
| Warm Savanna & Open Shrubland | WaSa |
| Warm Grassland | WaGr |
| Temperate Woody Savanna, Woodland & Shrubland | TeWo |
| Temperate Savanna & Open Shrubland | TeSa |
| Temperate Grassland | TeGr |
| Montane Grassland | MoGr |
| Arctic Tundra | ArTu |
| Desert | Des |
| Rocks and Ice | RoIc |
| Water | Wat |



**(a)**

cftfrac average 1988-2017 in %

**(b)**

cftfrac fractions (0-1) sum 1500-2017

**(c)**

cftfrac fractions (0-1) sum 1500-2017

**Figure A1. (a)** Average land use between 1988-2017 in % of grid cell area. **(b)** Total historical sum of yearly land use fractions [0-1] with exponential, and **(c)** linear legend.



**(a)** ecorisk no nitrogen

**(b)** current land use extent

**(c)** vegetation structure change

**(d)** local change

**(e)** global importance

**(f)** ecosystem balance

**Figure A2. (a)** Change in biochemical compositions computed by EcoRisk (as in Figure 3, but excluding Nitrogen variables) between 1550-1579 and 1985-2016. **(b-e)** EcoRisk components: vegetation structure change, local change, global importance, ecosystem balance.





**Figure A3. Biome internal difference distribution for the forest biomes** computed using EcoRisk between each cell of the biome with the average cell of the biome (for 1550-1579 states). Note that only very few cells are classified as BoND or TeBE - therefore their distribution appears very homogeneous. For biome names and abbreviations see Table A1.



|  | TrRF | TrDF | TeBD | TeNE | BoNE | BoBD | BoND | WaWo | WaSa | WaGr | TeWo | TeSa | TeGr | MoGr | ArTu | Des |
|---|---|---|---|---|---|---|---|---|---|---|---|---|---|---|---|---|
| Tropical Rain. | 0 | 0.23 | 0.7 | 0.53 | 0.87 | 0.88 | 0.92 | 0.75 | 0.95 | 0.97 | 0.91 | 0.95 | 0.97 | 0.97 | 0.96 | 0.99 |
| Tropical Decid. Forest | 0.39 | 0 | 0.78 | 0.75 | 0.87 | 0.87 | 0.91 | 0.81 | 0.95 | 0.97 | 0.91 | 0.95 | 0.97 | 0.97 | 0.96 | 0.99 |
| Temp. Broad. Decid. Forest | 0.66 | 0.65 | 0 | 0.47 | 0.62 | 0.78 | 0.84 | 0.64 | 0.88 | 0.86 | 0.57 | 0.71 | 0.89 | 0.93 | 0.92 | 0.98 |
| Temp. Needle. Ever. Forest | 0.65 | 0.62 | 0.76 | 0 | 0.77 | 0.85 | 0.89 | 0.89 | 0.95 | 0.97 | 0.88 | 0.93 | 0.96 | 0.96 | 0.96 | 0.99 |
| Bor. Needle. Ever. Forest | 0.67 | 0.67 | 0.26 | 0.51 | 0 | 0.68 | 0.86 | 0.59 | 0.58 | 0.61 | 0.33 | 0.49 | 0.76 | 0.91 | 0.93 | 0.97 |
| Bor. Broad. Decid. Forest | 0.7 | 0.7 | 0.51 | 0.64 | 0.53 | 0 | 0.47 | 0.77 | 0.8 | 0.72 | 0.57 | 0.64 | 0.79 | 0.52 | 0.61 | 0.98 |
| Bor. Needle. Decid. Forest | 0.76 | 0.76 | 0.65 | 0.74 | 0.67 | 0.57 | 0 | 0.67 | 0.73 | 0.7 | 0.56 | 0.63 | 0.78 | 0.6 | 0.63 | 1 |
| Warm Woodland | 0.76 | 0.77 | 0.75 | 0.8 | 0.79 | 0.88 | 0.82 | 0 | 0.69 | 0.85 | 0.66 | 0.75 | 0.89 | 0.88 | 0.87 | 0.99 |
| Warm Savanna | 0.89 | 0.9 | 0.77 | 0.87 | 0.75 | 0.79 | 0.82 | 0.34 | 0 | 0.68 | 0.54 | 0.54 | 0.87 | 0.77 | 0.8 | 0.98 |
| Warm Grassland | 0.93 | 0.94 | 0.77 | 0.93 | 0.82 | 0.83 | 0.85 | 0.66 | 0.52 | 0 | 0.52 | 0.32 | 0.24 | 0.72 | 0.76 | 0.78 |
| Temp. Woodland | 0.74 | 0.75 | 0.38 | 0.72 | 0.5 | 0.82 | 0.79 | 0.53 | 0.63 | 0.48 | 0 | 0.14 | 0.67 | 0.85 | 0.85 | 0.93 |
| Temp. Savanna | 0.87 | 0.88 | 0.65 | 0.87 | 0.71 | 0.84 | 0.85 | 0.62 | 0.59 | 0.36 | 0.26 | 0 | 0.59 | 0.79 | 0.81 | 0.88 |
| Temp. Grassland | 0.97 | 0.97 | 0.86 | 0.96 | 0.9 | 0.87 | 0.82 | 0.73 | 0.69 | 0.38 | 0.63 | 0.49 | 0 | 0.75 | 0.75 | 0.63 |
| Montane Grassland | 0.88 | 0.89 | 0.81 | 0.89 | 0.83 | 0.68 | 0.52 | 0.73 | 0.69 | 0.49 | 0.68 | 0.61 | 0.51 | 0 | 0.08 | 0.91 |
| Arctic Tundra | 0.87 | 0.87 | 0.8 | 0.88 | 0.82 | 0.7 | 0.58 | 0.75 | 0.75 | 0.64 | 0.68 | 0.65 | 0.5 | 0.12 | 0 | 0.88 |
| Desert | 0.99 | 0.99 | 0.98 | 0.99 | 0.99 | 0.96 | 0.99 | 0.93 | 0.91 | 0.72 | 0.91 | 0.83 | 0.6 | 0.93 | 0.91 | 0 |

**Figure A4.** Comparison of the state difference (EcoRisk) between biomes according to the average biome cell (y: reference state, x: scenario state) evaluated for the average state of 1550-1579. The colorcode is the same as in Figure A3. Since only very few cells are classified as TeBE, this biome is left out. Biomes RoIce and Wat do not host vegetation and are also left out. For biome names and abbreviations see Table A1.



**(a)**

leaching change in %

**(b)**

n_volatilization change in %

**Figure A5.** Change in **(a)** leaching and **(b)** nitrogen volatilization between 1550-1579 and 1985-2016





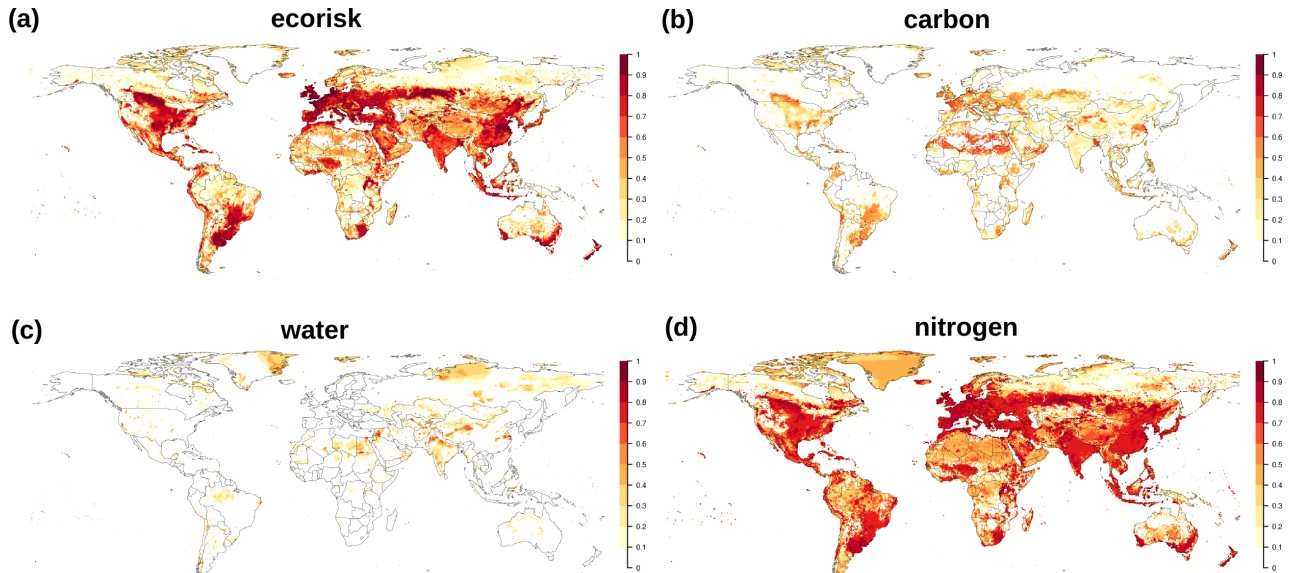

**Figure A6.** Change in biochemical compositions computed by EcoRisk between 1550-1579 and 1985-2016. **(a)** total ecorisk, and an evaluation of (gi+lc+eb)/3 only for components related to **(b)** carbon, **(c)** water, **(d)** nitrogen.



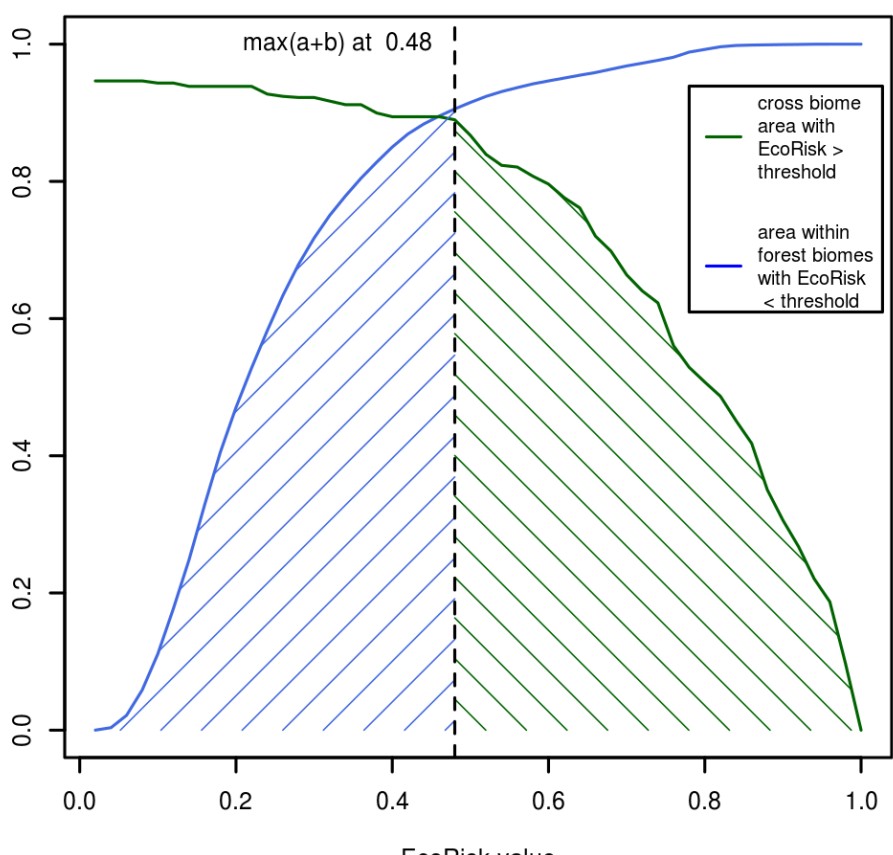

**Figure A7. Cumulative area** for all cells defined as not similar (Ecorisk > threshold) according to Figure A4 (green), and defined as similar (EcoRisk < threshold) to the biome average according to Figure A3 (blue). The value where the sum of both curves is maximal (an optimal threshold value fulfilling both criteria) is indicated by the dashed vertical line.



*Author contributions.*  FS: Conceptualization, Methodology, Software, Validation, Formal analysis, Writing - Original Draft, Visualization; JoB: Methodology, Software, Writing - Review & Editing; JaB: Software, Writing - Review & Editing; KE: Methodology, Writing - Review & Editing; DG: Writing - Review & Editing, Supervision; JH: Software (livestock-grassland calibration implementation); SM: Methodology, Data Curation, Writing - Review & Editing; SO: Methodology, Software, Writing - Review & Editing; SS: Conceptualization, Data Curation, Writing - Review & Editing; WL: Conceptualization, Writing - Review & Editing, Supervision, Funding acquisition

*Competing interests.*  The authors declare no competing interests.

*Acknowledgements.*  We thank the *Ecosystem in Transitions* group at PIK and especially Boris Sakschewski for fruitful discussions. FS is funded by the Global Challenges Foundation via Future Earth.




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
