# Peer review of "biospheremetrics v1.0.2: An R package to calculate two complementary terrestrial biosphere integrity indicators: human colonization of the biosphere (BioCol) and risk of ecosystem destabilization (EcoRisk)"

_EGUsphere, 2023_

## Author Response (AR1)

**Reviewer comment by Juan Rocha:**

**General comments:**
**Thanks for the opportunity to read your work. The manuscript is set to describe a new R package to calculate metrics of integrity and stability in ecosystems. The paper is clear and methods well explained with some simulation results coming from LPJmL models.**

**The paper, however, falls short in introducing the software, its requirements and documentation for a larger adoption by the community. If the central goal of the paper is to introduce the `biosphericmetrics` package, more attention should be given to the software documentation. For example, the package does not contain vignettes or demos that help the user understand what are the main functionalities. Since the package is meant to be used as companion of LPJmL model output, the authors should make available a small datasets that the users can test to understand the usage of functions (the equivalent of mtcars in other packages, not the 3GB in Zenodo). The user should be able to install the package, read the help of functions, and be able to run a demo with some results. I understand the output can be large files, but you can get inspiration of other packages such as stars or terra in making available small spatial data cubes (NetCDF files) for testing.**

**The GitHub repository has some useful information on for example what functions to use when working on small datasets, sending jobs to servers, or plotting. However, the package documentation is not stand alone. For example, it seems one needs json files to use `calc_biocol()`, but that dependency is not documented on the help files of the function. Most functions on the package do not have examples and are not working out of the box, many assumes the user has access to Fabian's folder structure. Thus I can install the package, but as a reviewer I cannot test if it works. The package also lacks testing and better documentation on scalability.**

**I strongly recommend to check the standards for software development in specialised journals such as Journal of Open Source Software, and the Journal of Statistical Software (JStatSoft). Both of them recommend the workflow and standards of the rOpenSci consortia: https://stats-devguide.ropensci.org**
**At minimum the package should contain a vignette explaining its functionality and a demo data that users can use to test its functionality.**

**Best regards,**
**Juan**

Reply:

Dear Juan,
thank you for your review and the positive evaluation of the manuscript.

We appreciate your comments on the package and have incorporated them in the next package version.

We have:
- added demo data for two example cells
- added unit tests based on this data for the main functions
- changed the example script to become a vignette and also used the demo data there to make the use of the functions more accessible
- went over all function parameters again and tried to better explain their use and structure

- included examples for all relevant functions

We have additionally incorporated most of the code comments received directly on github by Guido Kraemer (https://github.com/stenzelf/biospheremetrics/issues).

Best,
Fabian (on behalf of all authors)

**Specific comments:**
**Line 190: Unfinished sentence**
Reply: Thank you for spotting this, we have removed the sentence.

**Reviewer 2 comment:**

**This manuscript presents software that computes two measures of human impact on the terrestrial system, i.e. the human colonization of the biosphere and the risk of ecosystem destabilization.**
**To my opinion, this is a relevant manuscript, however, I must admit that I am not specialized on this topic. I am in favor in adding the nitrogen components into the assessment.**
**The preprint is very well written and well-documented, with nice figures and maps.**
**I only have few minor/textual comments.**

Reply:
We thank the reviewer for their positive evaluation of the manuscript. Adding nitrogen fluxes and pools was one major change that we did compared to the original Gamma metric – made possible by recent developments in the LPJmL model, which now includes nitrogen cycling.

**L4: "2" should be "two"; A number smaller or equal to ten should be written fully, if the number is not accompanied with physical units that refer to processes. Check also the reminder of the manuscript (i.e. L65 4/four components; L174 7/seven).**
Reply: We have replaced single digit numbers in lines 4, 65, and 174.

**L5: I should start with (i), and (ii) on line 6;**
Reply: We have added (i) and (ii) as suggested.

**L8-10: this sentence is to large and should be split in order to improve readability; the word "extraction" here should be explained here since I did not understand what the authors mean at this stage of the manuscript; Later on, it becomes clearer;**
Reply: We split up the sentence and added an explanation for modification and extraction, now reading:
"Applied to simulations with the dynamic global vegetation model LPJmL5 for 1500-2016, we find that presently (period 2007-2016), large regions show modification and extraction of >25% of the preindustrial potential net primary production. The modification (degradation) of NPP as a result of land-use change and extraction in terms of biomass removal (from e.g. harvest) leads to drastic alterations in key ecosystem properties, which suggests a high risk for ecosystem destabilization."

**L19 and elsewhere. References in the text: I am not familiar with the official formats of this journal, but to my opinion it is more logical to range the citations chronologically first and then alphabetically. After all, one refers first to the paper that came first;**
Reply: We have added references when possible in the order of relevance for the sentence. E.g. in line 19 the references refer to the points "carbon sequestration, moisture recycling, and resilience to disturbances/disruptions" in that order. We can rearrange this of course, if required by the journal.

**L51: Should be, "HANPP, however, cannot be …";**
Reply: We have changed the sentence to be:
"The original definition of $\Gamma$, however, did not include nitrogen variables and the code to compute it was hardly accessible to the scientific public. HANPP, on the other hand, so far was based on census statistics and inventory data and had not been calculated purely from vegetation model outputs."

**Add reference for Eq. 5:**
**Eqs 13-17 are referred from Ostberg et al. (2018)? If not, add reference;**
Reply: We have added Ostberg et al. (2018) as a reference for eq. 5.

**L190: insert whit space between 1000 and m:**
Reply: Done.

**L223-225: format "th" as superscript;**
Reply: Done, also in line 32.

**Fig 2: Use capital M for "map";**
Reply: Done.

**L245: Better write: "Values of gi are high in regions …";**
Reply: Done.

**L272-275: It would be better to use (i), (ii), (iii), when one lists up items;**
Reply: Done.

**L280: Better "Figure 5k,m";**
Reply: Done.

**L316-318: a list so use (i) and (ii);**
Reply: Done